# Comparative Study of Physical Activity, Leisure Preferences, and Sedentary Behavior among Portuguese, Italian, and Spanish University Students

**DOI:** 10.3390/healthcare12191930

**Published:** 2024-09-26

**Authors:** Rui Paulo, André Ramalho, Isabella Scursatone, Maria Caire, Nicolás Bores Calle, Daniel Bores-García, María Espada, Miguel Rebelo, Pedro Duarte-Mendes

**Affiliations:** 1Department of Sports and Well-Being, Polytechnic Institute of Castelo Branco, 6000-084 Castelo Branco, Portugal; andre.ramalho@ipcb.pt (A.R.); miguel.rebelo@ipcb.pt (M.R.); pedromendes@ipcb.pt (P.D.-M.); 2Sport Physical Activity and Health Research & INnovation CenTer, SPRINT, 2001-904 Santarém, Portugal; 3SUISM, Università degli Studi di Torino, 10126 Turin, Italy; isabella.scursatone@unito.it (I.S.); maria.caire@unito.it (M.C.); 4Department of Didactics of Body Expression, Faculty of Education of Palencia, Universidad de Valladolid, 47002 Valladolid, Spain; nicolasjulio.bores@uva.es; 5Department of Physiotherapy, Occupational Therapy, Rehabilitation and Physical Medicine, Faculty of Health Sciences, Universidad Rey Juan Carlos, 28032 Madrid, Spain; daniel.bores@urjc.es (D.B.-G.); maria.espada@urjc.es (M.E.)

**Keywords:** higher education students, sedentary lifestyle, physical activity, health

## Abstract

Objective: The objective of this study is to describe and compare the levels of physical activity, preferences for leisure-time physical activity, and the frequency of non-sedentary behaviors of Portuguese, Italian, and Spanish students attending higher education. Methods: A total of 1354 students (21.2 ± 2.9 years) participated in the study, with data collected through an online questionnaire for 6 months. Results: The highest levels of sedentary behavior are found among Spanish students, followed by the Portuguese, and lastly the Italians. In relation to physical activity levels, Spanish students perform more low and moderate physical activity, while Italian students perform more vigorous activities and naturally have a lower level of sedentary behavior. Conclusions: However, it is worth highlighting that students from all three countries reach the minimum levels of physical activity recommended by the WHO.

## 1. Introduction

University students face various academic, financial, and social challenges that can negatively affect their physical health, academic success, and quality of life [1]. The health benefits of physical activity (PA) are well-established and extensively documented in the scientific literature. PA is an important factor in the prevention and control of cardiovascular diseases [2], type 2 diabetes [3], obesity [4], and certain cancers [5]. Furthermore, PA can also produce positive mental health outcomes, including preventing cognitive decline [6] and reducing depressive symptoms and anxiety levels [7]. PA also improves muscle strength [8] and contributes to bone health [9].

However, despite the health benefits of daily PA, young Europeans are not physically active enough to benefit their health [10]. On the other hand, three or more hours per day of sedentary behavior was associated with an increased mortality risk, except in the most physically active individuals. In these individuals, an increased risk of mortality was found when they sat for five or more hours a day [11]. In addition, insufficient PA represents an urgent global public health challenge, affecting millions of people in developed and developing countries [12]. Therefore, physical inactivity is widely recognized as a major contributor to the development of numerous chronic diseases [13]. This inactivity is closely associated with an elevated risk of non-communicable diseases such as type 2 diabetes, cardiovascular disease, several major cancers, musculoskeletal disorders, and a wide spectrum of other adverse health outcomes in different population segments [14]. Extensive evidence shows that insufficient PA impacts almost every cell, organ, and system in the body, significantly increasing the risk of premature mortality [11]. In fact, it is estimated that reducing overall levels of physical inactivity could increase average life expectancy by 0.68 years [15]. Addressing this issue requires a concerted effort to promote active lifestyles worldwide, given the profound impact of physical inactivity on public health.

Healthy lifestyle habits are generally recognized as the integration of behaviors such as maintaining a balanced diet, practicing regular PA, ensuring adequate rest, consuming alcohol responsibly, and avoiding drug use [16]. In addition to their positive impact on physical health, these habits also play a key role in improving emotional well-being in university students [17]. This link underlines the importance of promoting the adoption and maintenance of healthy habits as a key strategy for improving the health outcomes of this population subgroup.

Previous studies have assessed the physical activity (PA) levels of Polish university students, both through self-reported measures [18] and accelerometry [19]. Other research has examined factors influencing PA and sedentary behavior among Spanish university students [20], and the association between vigorous PA and various psycho-social variables [21]. Additionally, some studies have explored the impact of sports participation on the quality of life of student–athletes [22]. Investigations have also objectively measured sedentary behavior and PA levels in university students examining their relationship with body mass index [23]. This study seeks to address the current gap in the literature by offering comprehensive data on the PA levels of university students in Portugal, Spain, and Italy, where such data remain limited, especially in comparative studies across these three countries.

It is important to remember that PA and sedentary behavior can be described through many activities carried out in multiple contexts and, potentially, with different determining factors and health outcomes [24,25]. In this sense, it is also essential to know population trends when choosing these activities [26]. All adults are encouraged to engage in regular physical activity for optimal health. Specifically, they should aim for 150–300 min of moderate-intensity aerobic activity or 75–150 min of vigorous-intensity aerobic activity weekly, or a combination of both intensities. Additionally, adults should incorporate muscle-strengthening exercises targeting all major muscle groups on at least two days per week, as these activities offer further health benefits. This balanced approach promotes substantial improvements in physical and overall well-being [26].

This study aims to describe and compare the physical activity levels of students in Portugal, Italy, and Spain, focusing on those attending higher education institutions in each respective country. By providing new insights into the PA behaviors of students in these regions, this study contributes to a better understanding of their health-related habits, which are essential for informing targeted interventions and promoting active lifestyles within higher education institutions. Based on the existing literature and our own experience, the central hypothesis of this study is that higher education students, while often complying with international PA guidelines, are still predominantly engaged in sedentary behaviors. Furthermore, comparisons between countries may reveal variations, potentially attributable to different national policies and levels of investment in promoting active lifestyles. 

## 2. Methods

This is a cross-sectional study, carried out in 3 countries (Portugal, Spain, and Italy), based on epidemiological studies [27]. We used the quantitative method [28], which uses statistical techniques to quantify data collection and processing. This is a collaborative study between the authors of these countries and with the same database, and with data collected on the same dates (data collection lasted 6 months).

### 2.1. Participants

A non-probabilistic sample of Portuguese, Italian, and Spanish students was selected through a sampling strategy in two phases. The first stage of the sampling process was based on the selection of Portuguese, Spanish, and Italian higher education institutions stratified by the regions of the three countries. The second stage of the sampling strategy was based on the selection of students enrolled in different higher education institutions, according to the different scientific areas of higher education courses. The selection of participants was carried out in successive recruitment phases in order to use the updated lists of students enrolled in the different courses of the three different countries. In this sense, the aim was to recruit students with Portuguese, Spanish, and Italian nationality, aged 18 or over, who attend higher education courses from different scientific areas taught by public higher education institutions, such as universities and polytechnics institutes, from different regions of Portugal, Spain, and Italy.

Thirteen hundred and fifty-four (1354) Portuguese, Italian, and Spanish students who attended universities in Portugal, Italy, and Spain participated in this study (Table 1). Of this total, 385 subjects studied in higher education institutions in Portugal (average age 20.9 ± 2.9), 398 studied in higher education institutions in Italy (average age 22.3 ± 2.7), and 571 studied in higher education institutions in Spain (average age 20.6 ± 2.9).

### 2.2. Instruments

The short version of the International Physical Activity Questionnaire (IPAQ) was applied, validated, and translated into Italian [29], Portuguese [30], and Spanish [31]. The questionnaires were sent via an online form (through e-mail), and each of the validated versions for each country was administered to students in the respective country. This instrument makes it possible to standardize measures related to health and the assessment of physical activity behaviors of the population in different countries and in different sociocultural contexts [32]. The short version of the IPAQ was used because it is a questionnaire that is easier, faster, and more viable to complete in studies involving a large number of participants [30]. Using the IPAQ scoring protocol, it was possible to estimate the total weekly physical activity by evaluating the time spent at each intensity of activity with its estimated metabolic equivalent energy expenditure [30]. According to the IPAQ results, students can be classified as low active, moderately active, or highly active. 

Moderately active means individuals achieved at least 600 metabolic equivalent minutes per week. High means that individuals achieved at least 3000 metabolic equivalent minutes per week. Low activity indicates that individuals did not meet the “moderately” or “high” criteria. Participants in the present study also answered whether they regularly engage in physical activities in their free time. If the answer was “Yes,” they listed these activities and the weekly frequency and number of minutes per day that they dedicate to carrying out each physical activity mentioned.

### 2.3. Procedures

Formal and institutional contact was made with the higher education institutions, presenting the study’s objectives and requesting authorization. Before data collection, all subjects were presented with the study in question, its objectives, and the procedures to be followed. An anamnesis form and an informed consent form were sent to each subject to conduct the evaluations, with all ethical principles, international norms, and standards related to the Declaration of Helsinki and the Convention on Human Rights being respected and preserved [33]. All assessments were carried out by sending questionnaires via email to the subjects. Although the sample was non-probabilistic, the selection of participants, after institutional contact, was performed randomly, to minimize the influence of confounding variables. The study was approved by the ethics committee (opinion no. 58 CE-IPCB/2021).

### 2.4. Statistical Analyses

#### 2.4.1. Preliminary Analysis

An inspection of the data revealed no missing values or univariate outliers. An a priori power analysis through G*Power (3.1.9.7) and a one-way ANOVA was used as an alternative for the Kruskal–Wallis test as a non-parametric test [34] to determine the required sample size considering the following input parameters: effect size f = 0.25; α err prob = 0.01; statistical power = 0.95. The required sample size was 1008 (336 for each group), which was respected in the present study.

#### 2.4.2. Main Statistical Analysis

Descriptive statistics were performed for all analyzed variables, including mean and standard deviation. Then, a Kolmogorov–Smirnov (n > 50) test was performed to analyze the data distribution, considering *p* > 0.05 as a normal distribution [35]. All data variables analyzed presented with a non-normal distribution. A Kruskal–Wallis test was used to verify differences between groups. Moreover, a post hoc pairwise comparison was performed for groups that presented with statistical differences. Finally, an effect size (Cohen d) analysis was used to determine the magnitude of the effect, and the following cut-off values were considered: <0.2, trivial; 0.21–0.6, small; 0.61–1.2, moderate; 1.21–2.0, big; and >2.0, very big [36]. The effect size was calculated using the eta square value (η^2^) [37]. All statistical analyses were performed using the SPSS software v. 29.0 (IBM, Chicago, IL, USA), and the significance level was set at *p* ≤ 0.05 to reject the null hypothesis [35,38].

## 3. Results

Table 2 shows the differences in the studied variables between the three groups (Portugal, Spain, and Italy) regarding physical activity and sedentary time. Differences between groups (*p* ≤ 0.05) were found in all variables studied. 

Regarding post hoc pairwise comparisons for groups, differences were found in all variables studied (*p* ≤ 0.05), except moderate physical activity (*p* = 0.525) and vigorous physical activity (*p* = 0.075) between Portugal and Spain. Figure 1 shows variance analyses of the time spent on different behaviors in the three countries studied. Regarding sitting (weekday/day), we found differences between Portugal and Spain (*p* < 0.001, η^2^ = 0.164, effect size = 0.885), Portugal and Italy (*p* < 0.001, η^2^ = 0.074, effect size = 0.565), and Spain and Italy (*p* < 0.001, η^2^ = 0.236, effect size = 1.113). Further differences were found in the following variables: sitting (weekend/day) between Portugal and Spain (*p* < 0.001, η^2^ = 0.064, effect size = 0.524), Portugal and Italy (*p* < 0.001, η^2^ = 0.142, effect size = 0.813), and Spain and Italy (*p* < 0.001, η^2^ = 0.205, effect size = 1.014); walking between Portugal and Spain (*p* < 0.001, η^2^ = 0.146, effect size = 0.828), Portugal and Italy (*p* = 0.011, η^2^ = 0.054, effect size = 0.476), and Spain and Italy (*p* < 0.001, η^2^ = 0.091, effect size = 0.633); moderate activity between Portugal and Italy (*p* < 0.001, η^2^ = 0.074, effect size = 0.567) and Spain and Italy (*p* = 0.011, η^2^ = 0.049, effect size = 0.453); and vigorous activity between Portugal and Italy (*p* < 0.001, η^2^ = 0.118, effect size = 0.732) and Spain and Italy (*p* < 0.001, η^2^ = 0.076, effect size = 0.573).

Figure 2 shows the variance analyses related to the metabolic equivalent of the task in each country. We found differences between Portugal and Spain (*p* < 0.001, η^2^ = 0.021, effect size = 0.293) and Portugal and Italy (*p* < 0.001, η^2^ = 0.051, effect size = 0.461).

## 4. Discussion

The objective of the present research was to describe and compare the levels of physical activity, preferences for leisure-time physical activity, and frequency of non-sedentary behaviors of Portuguese, Italian, and Spanish students attending higher education in Portugal, Italy, and Spain, respectively.

The findings of the present study show an interaction effect between country and sedentary behavior, with the highest levels of sedentary lifestyles found among Spanish students, followed by the Portuguese, and, lastly, the Italians. 

Sedentary individuals usually present a greater number of barriers, whilst active individuals present a lower number of barriers. Previous studies show that the main barrier for Spanish university students is that they do not like to practice physical activity. This barrier has a high negative correlation with the levels of physical activity [39,40]. A recent study shows that less than half of Spanish university students can be considered physically active [41].

However, people studying in Portugal spent the most time sitting during one working day compared with Polish and Belarusian students [42]. These data are different to those collected in the Eurobarometer 2022 [10] in which 40% of Italian citizens claimed to spend at least 330 min per day (five and a half hours) doing sedentary activities, 34% in the case of the Spanish, and 29% in the case of the Portuguese. These data are quite worrisome since an increased mortality risk was found when they sat for five or more hours a day [11]. 

Regarding physical activity levels, Spanish students perform more minutes of low and moderate activity than Italian and Portuguese students. Italians are the ones who perform more vigorous activity. It is also observed that Portuguese people practice the least number of minutes of physical activity at all levels. According to the World Health Organization (WHO), adults engage in at least 150 min per week of moderate-intensity aerobic physical activity, or at least 75 min of vigorous-intensity aerobic physical activity throughout the week, or an equivalent combination of moderate- and vigorous-intensity activities [43]. The observed differences in physical activity levels may be due to factors such as cultural differences and different mechanisms of inclusion of physical activity in education systems [44]. According to Burton et al. [45], university students typically report low levels of physical activity due, among other factors, to a greater demand on time for study. A study conducted in Spain with 3060 university students concluded that more than 60% of the students did not meet the minimum levels of physical activity recommended by the WHO [41]; in other study, only 58% of Spanish students reached the recommendations established by the WHO [46]. Another research study conducted in Poland, Portugal, and Belarus with a total of 1136 university students showed that the dominant level of activity was at a vigorous physical activity level in 58.5% of the surveyed men and 46.2% of the surveyed women [42]. However, the results show a higher compliance with WHO recommendations compared to the results obtained in a recent systematic review with a meta-analysis carried out in 32 countries, where the percentage of adherence to these international recommendations is 17.12% in adults (which is the population sector in which university students are found) and 19.74% in adolescents [47]. Another previous study also points, in line with the results presented in this paper, to an attainment of the required levels of physical activity among university students [48]. In this line of research, the study conducted by Dabrowska-Galas et al. [18] with a total of 300 university students found that 98% of the physical therapy students claimed that they were physically active before starting university. In the same way, an insufficient level was recorded among 12.6% of male university students and 16.7% of female university students from Poland, Portugal, and Belarus [42].

### Limitations and Strengths

The main limitation of this study lies in its comparison of only three countries in Europe. There are also other limitations such as the use of only subjective instruments (IPAQ) for data collection and not having used other instruments that allow us to analyze other social, motivational, and cultural factors to see what effect they may have on the levels of physical activity.

However, it should be noted that IPAQ is justified as a strength in this research study because it shows validity. It has been used in national and international studies.

We present, as suggestions for future investigations, the possibility of using less subjective instruments, which can provide more objective data and lead to more specific conclusions. In addition to the indicators assessed, eating habits and some body composition indicators can also be assessed to complement these analyses. Also, there is the possibility of involving more countries in comparative studies of this nature.

Regarding the practical implications of this study, we consider it to be another modest contribution to demonstrating the evidence pointing to the sense that the young adult population (higher education students) presents with worryingly low values of physical activity and worryingly high values concerning sedentary behaviors. This could be another contribution so that public policies can direct greater attention and investment to this problem, which is emerging, as well as universities being able to rethink their internal organization around enabling their students to spend more time dedicated to active behaviors and minimize sedentary behaviors, both in teaching activities and in other academic activities. The promotion of physical sports activities that help to increase the weekly physical activity time, even if at the beginning it still does not reach the minimum required, can be a good start and, undoubtedly, much better than the scarce physical activity shown in the results.

## 5. Conclusions

Spanish students engage in more low and moderate intensity exercise, while Italian students spend more time in vigorous physical activity. All three populations meet the WHO’s minimum recommended activity levels. This study finds that Spanish students are the most sedentary, followed by Portuguese, and then Italian students. The analysis between weekdays and weekends shows that Italian students are consistently the least sedentary. High sedentary behavior is likely linked to the nature of university life, involving extensive sitting for lectures and study. It is crucial for universities to create opportunities for students to increase physical activity, countering sedentary time and promoting long-term healthy lifestyles.

## Figures and Tables

**Figure 1 healthcare-12-01930-f001:**
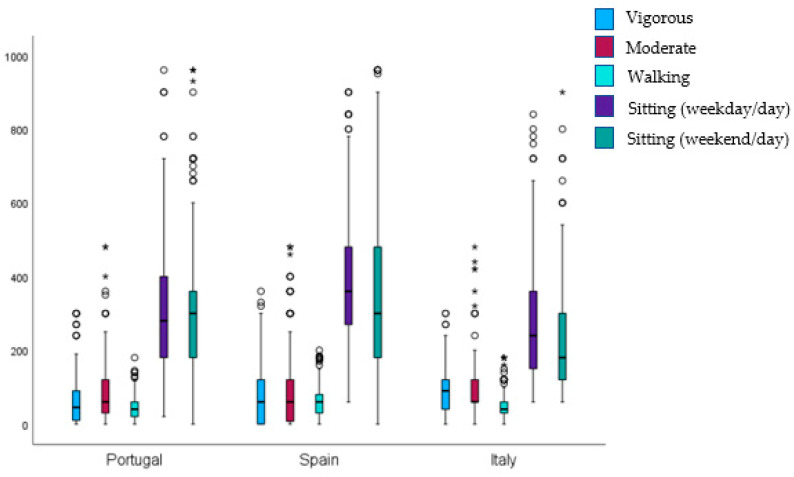
Time spent in the different types of behaviors analyzed (minutes/day). (Note: * values that are significantly higher than the interquartile range).

**Figure 2 healthcare-12-01930-f002:**
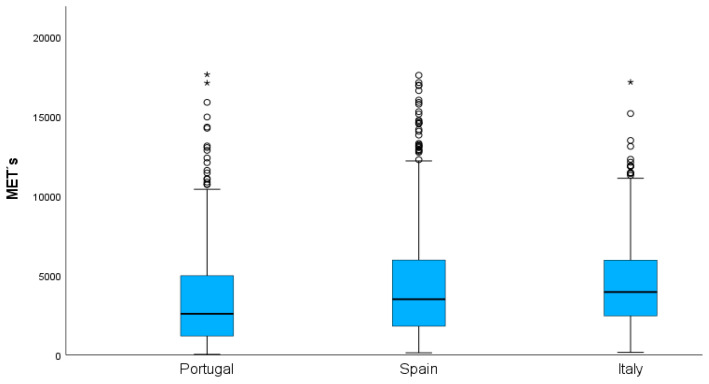
Metabolic equivalent of task in each country. (Note: * values that are significantly higher than the interquartile range).

**Table 1 healthcare-12-01930-t001:** Sample characteristics regarding gender, age, and degree.

Characteristics	Total	Portugal	Spain	Italy
N (%)	1354 (100)	385 (28)	571 (43)	398 (29)
Gender, N (%), (IC95%)				
Female	815 (60.2) (57.5–62.9)	227 (59) (51.4–61.0)	409 (71.6) (68.3–75.5)	189 (47.5) (42.7–52.3)
Male	530 (39.1) (36.5–41.9)	164 (42.6) (37.9–47.8)	158 (27.7) (23.8–31.2)	208 (52.3) (47.5–57.0)
Diverse	9 (0.7) (0.3–1.1)	4 (1.0) (0.3–2.1)	4 (0.7) (0.2–1.6)	1 (0.2) (0–0.8)
Age, range, years (mean ± SD, median)	17–35 (21.2 ± 2.9; 21.0)	17–35 (20.9 ± 2.9; 20.0)	17–35 (20.6 ± 2.9; 20.0)	19–34 (22.3 ± 2.7; 22.0)
Degree, N (%), (IC95%)				
Bachelor	1237 (91.4) (89.9–92.8)	317 (82.3) (78.7–86.0)	529 (92.6) (90.5–94.9)	391 (98.2) (97–99.2)
Master	78 (5.8) (4.4–7.0)	39 (10.1) (7.3–13.0)	37 (6.5) (4.4–8.6)	2 (0.5) (0–1.3)
Other	39 (2.9) (2.0–3.8)	29 (7.5) (5.2–10.1)	5 (0.9) (0.2–1.8)	5 (1.3) (0.3–2.5)
Sub-elite	39	25.36 ± 4.83	72.56 ± 7.99	1.73 ± 0.05
Amateur	16	22.01 ± 3.55	72.96 ± 15.61	1.76 ± 0.07

**Table 2 healthcare-12-01930-t002:** Differences between groups regarding physical activity and sedentary time (min/d).

	Groups	N	M	M CI 95%	SD	*p*	η^2^	Effect Size
Sitting (weekday/day)	Portugal	385	300.71	283.35–318.08	173.32	<0.001	0.127	0.762 (moderate)
	Spain	571	389.26	376.01–402.51	161.20			
	Italy	398	258.64	244.21–273.07	146.42			
Sitting (weekend/day)	Portugal	385	298.57	279.83–317.31	189.00	<0.001	0.088	0.623 (moderate)
	Spain	571	343.65	326.68–360.63	206.53			
	Italy	398	212.54	199.28–225.80	134.57			
Walking	Portugal	385	44.73	47.78–42.23	30.47	<0.001	0.045	0.433 (small)
	Spain	571	61.53	58.39–64.67	38.18			
	Italy	398	51.53	47.93–55.12	36.52			
Moderate	Portugal	385	79.48	71.99–86.97	74.76	0.001	0.008	0.183 (trivial)
	Spain	571	91.59	83.93–99.26	93.30			
	Italy	398	89.74	83.50–95.99	63.40			
Vigorous	Portugal	385	62.06	55.70–68.43	63.49	<0.001	0.023	0.309 (small)
	Spain	571	79.60	72.60–86.61	85.21			
	Italy	398	80.82	75.57–86.06	53.21			

*p* ≤ 0.05—Kruskal–Wallis test significance level. N, number of subjects; M, mean; M CI, confidence interval for the mean values; SD, standard deviation; η2, eta square value.

## Data Availability

The data presented in this study are only available upon request from the corresponding author. The data are not publicly available due to privacy issues.

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
