# Peer review of "Comparative Study of Physical Activity, Leisure Preferences, and Sedentary Behavior among Portuguese, Italian, and Spanish University Students"

_healthcare, 2024, doi:10.3390/healthcare12191930_

Round 1
Reviewer 1 Report
Comments and Suggestions for Authors
Review Report (healthcare-3169029)
Dear Authors,
Congratulations on the manuscript entitled “Self-reported Physical Activity Levels, Leisure-time Physical Activity Preferences and Frequency of Non-Sedentary Behaviors: Comparative Study between Portuguese, Italian and Spanish Students Attending Higher Education” submitted to the journal Healthcare. The comparison between university students from Portugal, Italy and Spain regarding physical activity levels, leisure-time physical activity preferences and non-sedentary behaviors is interesting. Despite the self-reported data, justified by the nature of the research, the study is relevant in the context of public health promotion. However, the manuscript needs improvements in terms of writing, organization and references (17/38 references are over 10 years old). I leave my comments below so that you can reflect on improvements that I consider important. My opinion has been forwarded to the Editor.
Best regards,
------------------------------------------------------------------
Title and abstract
- The title is not attractive and should be simplified. I suggest “Comparative Study of Physical Activity, Leisure Preferences, and Sedentary Behavior among Portuguese, Italian, and Spanish University Students”.
- You need to pay attention to the quality of the writing throughout the text. In the abstract, there is the expression (1) that seems completely out of context. Please review the abstract according to the instructions in the Healthcare template.
- The repetition of "Portugal, Italy and Spain" should be used sparingly. Please review.
- The description of the statistical procedure and data collection seems confusing. I recommend omitting the information on the sample power and focusing on information about the variables.
- Use consistent terms, for example, "sedentary behavior" instead of "sedentary lifestyles".
Introduction
- Adjust the text to be more direct by deleting the followin sentence “However, physical activity can be an important ally in improving students' health. The health benefits of physical activity are recognized and consolidated in scientific literature.”.
- The word “physical activity” appears 67 times throughout the manuscript and should be abreviated to “PA”.
- Review all references. For example: Reference [9] on osteoporosis is old and was based on a study with rats. Thre are studies with humans that make the same inference.
- The second paragraph provides a detailed presentation of several studies and may be more direct, thus leaving the reader confused about where the authors are actually going. Adjust the second paragraph to be more concise by reducing the details of previous research.
- Please clarify “specific physical activity behaviors” in: “Despite these important contributions, a limitation of most previous studies is the failure to characterize the specific physical activity behaviors that university students prefer or habitually engage in. Furthermore, to the best of our knowledge, the study of non-sedentary behavior of university students also appears to be scarce.”
- I confess that I read the introduction several times to understand what the focus/gap of the work was. I was unable to identify a specific problem that the study seeks to solve. I recommend that the authors review this point.
Methods
- The description presented in lines 77 to 81 is unnecessary. Just inform that this is a cross-sectional study conducted in the respective countries and the years of collection. The authors can also inform if the study is collaborative with authors from these same countries and if all the data are in the same database. If the data were collected in different years, it will also be necessary to mention this.
- Add information about how the institutions and students were recruited.
- Please explain the physical activity classification categories (low, moderate, high) in the instruments section, so that the reader does not need to look for this information outside the manuscript.
- Was there any type of control for confounding variables, such as gender, socioeconomic status or health history? How were they considered in the analyses?
- The authors declared that there were missing values ​​or univariate outliers. But how did they identify outliers and how was the robustness of the analyses ensured in the absence of outliers?
- If the data did not show normalityty, why mention the use of ANOVA? Could this be based on the central limit theorem theory?
Result
Although the results are interesting, the authors need to reflect on what they actually intend to achieve with the study. The differences found between countries are clear, but the connection between these results and the study objectives is not (which involves describing and comparing physical activity levels, preferences and non sedentary behaviors). How do these results contribute to the understanding of preferences and non-sedentary behaviors of university students? In my opinion, the study objective needs to be revised, since no information on preferences was collected. Practicing more time in PA or SB does not mean that these are the preferred activities of university students. Please consider moderating the study objective.
Discussion
Overall, the discussion is based on the interpretation of the results, which sometimes seems confusing. For example: For example, the authors mention that the data are different from those collected in the Eurobarometer 2022, but do not explore possible reasons for these differences, such as methodological variations, cultural differences, or specificities of the sample studied. This is repeated for all findings.
- The first paragraph should primarily highlight the main findings of the study in line with its objectives, and present the practical implications for health care.
- Lines 172 to 180. The authors only compared the results with previous studies to report differences at the country level. But what explains the differences in sedentary behavior among Portuguese, Italian, and Spanish university students?
- Third and fourth paragraphs: The previous comment also applies to the differences observed in physical activity among university students in the three countries.
- The authors need to include limitations of the study. Include the reliance on subjective instruments (IPAQ) for data collection, the possible unrepresentativeness of the sample, or the lack of data from othr European countries. Furthermore, if the data were collected at differents times in the three countries, this may introduce seasonal (seasonal) or temporal bias that was not controlled for in the study. The lack of informations on cultural difference may also limit the interpretation of the results. Also include the lack of data on othr factors that may influence physical activity levels, such as the built environment, social support, and individual motivation, which limit the understanding of the results.
Conclusions
- The authors do not mention preferences for physical activities, which were part of the study objectives.
Author Response
Review Report (Reviewer 1)
Title and abstract
Reviewer Comments:
- The title is not attractive and should be simplified. I suggest “Comparative Study of Physical Activity, Leisure Preferences, and Sedentary Behavior among Portuguese, Italian, and Spanish University Students”.
- You need to pay attention to the quality of the writing throughout the text. In the abstract, there is the expression (1) that seems completely out of context. Please review the abstract according to the instructions in the Healthcare template.
- The repetition of "Portugal, Italy and Spain" should be used sparingly. Please review.
- The description of the statistical procedure and data collection seems confusing. I recommend omitting the information on the sample power and focusing on information about the variables.
- Use consistent terms, for example, "sedentary behavior" instead of "sedentary lifestyles".
Answer: Thanks for your comment. As requested, we changed the title of the manuscript and reformulated the entire abstract.
Review:
Title: ”Comparative Study of Physical Activity, Leisure Preferences, and Sedentary Behavior among Portuguese, Italian, and Span-ish University Students”.
Abstract: “The objective of this study is to describe and compare the levels of physical activity, preferences for leisure-time physical activity and the frequency of non-sedentary behaviors of Portuguese, Italian and Spanish students attending higher education; 1354 students (21.2 ± 2.9 years) participated in the study participated in the study with data collected through an online questionnaire for 6 months; The highest levels of sedentary behavior are found among Spanish students, followed by the Portuguese and lastly the Italians. In relation to physical activity levels, Spanish students are those who perform more low and moderate physical activity, while Italian students are those who perform more vigorous activities and naturally have a lower level of sedentary behavior. However, it is worth highlighting that all three countries reach the minimum levels of physical activity recommended by the WHO”.
Page number 1, Line number 15-24.
Introduction
Reviewer Comment: Adjust the text to be more direct by deleting the followin sentence “However, physical activity can be an important ally in improving students' health. The health benefits of physical activity are recognized and consolidated in scientific literature.”.
Answer: (ll.34-36). The health benefits of physical activity (PA) are well-established and extensively documented in scientific literature.
Reviewer Comment: The word “physical activity” appears 67 times throughout the manuscript and should be abreviated to “PA”.
Review all references. For example: Reference [9] on osteoporosis is old and was based on a study with rats. Thre are studies with humans that make the same inference.
Answer: (ll.40). The reference was replaced by Carter, M.I.; Hinton, P.S. Physical Activity and Bone Health. Mo. Med. 2014, 111, 59–64.
Reviewer Comment: The second paragraph provides a detailed presentation of several studies and may be more direct, thus leaving the reader confused about where the authors are actually going. Adjust the second paragraph to be more concise by reducing the details of previous research.
Answer: (ll.65-75). Previous studies have assessed the physical activity (PA) levels of Polish university students, both through self-reported measures [18] and accelerometry [19]. Other research has examined factors influencing PA and sedentary behavior among Spanish university students [20], and the association between vigorous PA and various psychosocial variables [21]. Additionally, some studies have explored the impact of sports participation on the quality of life of student-athletes [22]. Investigations have also objectively measured sedentary behavior and PA levels in university students examining their relationship with body mass index [23]. This study seeks to address the current gap in the literature by offering comprehensive data on the PA levels of university students in Portugal, Spain, and Italy, where such data remains limited, especially in comparative studies across these three countries.
Reviewer Comment: Please clarify “specific physical activity behaviors” in: “Despite these important contributions, a limitation of most previous studies is the failure to characterize the specific physical activity behaviors that university students prefer or habitually engage in. Furthermore, to the best of our knowledge, the study of non-sedentary behavior of university students also appears to be scarce.”
Answer: (ll.72-75). This sentence there was replaced for this: “This study seeks to address the current gap in the literature by offering comprehensive data on the PA levels of university students in Portugal, Spain, and Italy, where such data remains limited, especially in comparative studies across these three countries.”
Reviewer Comment: I confess that I read the introduction several times to understand what the focus/gap of the work was. I was unable to identify a specific problem that the study seeks to solve. I recommend that the authors review this point.
Answer: (ll.72-75). This study seeks to address the current gap in the literature by offering comprehensive data on the PA levels of university students in Portugal, Spain, and Italy, where such data remains limited, especially in comparative studies across these three countries.
Action (ll.79-85). This study aims to describe and compare the physical activity levels of students in Portugal, Italy, and Spain, focusing on those attending higher education institutions in each respective country. By providing new insights into the PA behaviors of students in these regions, this study contributes to a better understanding of their health-related habits, which are essential for informing targeted interventions and promoting active lifestyles within higher education institutions.
Methods
Reviewer Comment: The description presented in lines 77 to 81 is unnecessary. Just inform that this is a cross-sectional study conducted in the respective countries and the years of collection. The authors can also inform if the study is collaborative with authors from these same countries and if all the data are in the same database. If the data were collected in different years, it will also be necessary to mention this.
Answer: Thanks for your comment. As requested, we make the changes proposed by the reviewer. “This is a cross-sectional study, carried out in 3 countries (Portugal, Spain and Italy), based on epidemiological studies [21]. We used the quantitative method [22], which uses statistical techniques to quantify data collec-tion and processing. This is a collaborative study between the authors of these coun-tries and with the same database, and with data collected on the same dates (data collection lasted 6 months).” Page 2, Line number 77-83.
Reviewer Comment: Add information about how the institutions and students were recruited.
Answer: Thanks for your comment. This information can be found in the Procedures section: “Formal and institutional contact was made with Higher Education institutions, presenting the study's objectives and requesting authorization. Before data collection, all subjects were presented with the study in question, its objectives, and the procedures to be followed” Page number 3, Line number 122-125.
Reviewer Comment: Please explain the physical activity classification categories (low, moderate, high) in the instruments section, so that the reader does not need to look for this information outside the manuscript.
Answer: Thanks for your comment. This information can be found in the Instruments section: “Moderately active means individuals achieved at least 600 metabolic equivalent minutes per week. High means that individuals achieved at least 3000 metabolic equivalent minutes per week. Low activity indicates that individuals do not meet the “moderately” or “high” criteria.” Page number 3, Line number 115-118.
Reviewer Comment: Was there any type of control for confounding variables, such as gender, socioeconomic status or health history? How were they considered in the analyses?
Answer: Thanks for your comment. As requested, we make the changes proposed by the reviewer. “Although the sample was non-probabilistic, the selection of participants, after institutional contacts, was done randomly, to minimize the influence of confounding variables.” Page number 4, Line number 130-132.
Reviewer Comment: The authors declared that there were missing values ​​or univariate outliers. But how did they identify outliers and how was the robustness of the analyses ensured in the absence of outliers?
Answer: Thanks for your comment. The verification of outliers was carried out using the Boxplot in SPSS software version 29.0, through the Tukey method interquartile range to detect values that deviate significantly from the central portion of the data. (George, D., & Mallery, P. (2024) IBM SPSS Statistics 29 Step by Step: A Simple Guide and Reference. (18th Ed.). New York and London: Routledge. doi:10.4324/9781032622156)
Reviewer Comment: If the data did not show normalityty, why mention the use of ANOVA? Could this be based on the central limit theorem theory?
Answer: Thanks for your comment. This information can be found in the Statistical Analyses (Preliminary Analysis) section. We only use ANOVA in the preliminary analysis, as an alternative, to determine the required sample size (Faul, et al., 2007): “and one-way ANOVA was used as an alternative for the Kruskal-Wallis test as a non-parametric test [28] to determine the required sample size” Page number 4, Line number 150-152.
Result
Reviewer Comment: Although the results are interesting, the authors need to reflect on what they actually intend to achieve with the study. The differences found between countries are clear, but the connection between these results and the study objectives is not (which involves describing and comparing physical activity levels, preferences and non sedentary behaviors). How do these results contribute to the understanding of preferences and non-sedentary behaviors of university students? In my opinion, the study objective needs to be revised, since no information on preferences was collected. Practicing more time in PA or SB does not mean that these are the preferred activities of university students. Please consider moderating the study objective.
Answer: Thanks for your comment. As requested, we make the changes proposed by the reviewer. Several changes were made to the introduction, as previously reported, including the study objectives.
Action (ll.79-85). This study aims to describe and compare the physical activity levels of students in Portugal, Italy, and Spain, focusing on those attending higher education institutions in each respective country. By providing new insights into the PA behaviors of students in these regions, this study contributes to a better understanding of their health-related habits, which are essential for informing targeted interventions and promoting active lifestyles within higher education institutions.
Discussion
Reviewer comment: Overall, the discussion is based on the interpretation of the results, which sometimes seems confusing. For example: For example, the authors mention that the data are different from those collected in the Eurobarometer 2022, but do not explore possible reasons for these differences, such as methodological variations, cultural differences, or specificities of the sample studied. This is repeated for all findings.
Answer: Thanks for your comment. As requested, we make the changes proposed by the reviewer. “The observed differences in physical activity levels may be due to factors such as cultural differences and different mechanisms of inclusion of physical activity in education systems” (with reference, lines 198-200).
Reviewer comment: Lines 172 to 180. The authors only compared the results with previous studies to report differences at the country level. But what explains the differences in sedentary behavior among Portuguese, Italian, and Spanish university students?
Answer: Thanks for the suggestion. We believe that with the addition presented in the previous suggestion we respond to it.
Reviewer comment: The authors need to include limitations of the study. Include the reliance on subjective instruments (IPAQ) for data collection, the possible unrepresentativeness of the sample, or the lack of data from othr European countries. Furthermore, if the data were collected at differents times in the three countries, this may introduce seasonal (seasonal) or temporal bias that was not controlled for in the study. The lack of informations on cultural difference may also limit the interpretation of the results. Also include the lack of data on othr factors that may influence physical activity levels, such as the built environment, social support, and individual motivation, which limit the understanding of the results.
Answer: Thanks for your comment. As requested, we make the changes proposed by the reviewer. “The main limitation of this study lies in compare only three countries of Europe. There are also other limitations such as use only a subjective instruments (IPAQ) for data collection and not having used other instruments that allow us to analyze other social, motivational and cultural factors to see what effect they may have on the level of physical activity.
However, it should be noted that IPAQ is justified as a strength in this research because show validity and reability. It has been used in national and international studies.” (lines 221-228)
Conclusions
Reviewer comment: The authors do not mention preferences for physical activities, which were part of the study objectives.
Answer: Thanks for your comment. As requested, we make the changes proposed by the reviewer. Several changes were made to the introduction, as previously reported, including the study objectives.

Reviewer 2 Report
Comments and Suggestions for Authors
The abstract is properly constructed
introduction
Prior research on physical activity and sedentary behavior was analyzed, but it is judged by the analysis of college students. In this section, I believe that the development of the introduction can be much smoother if I talk about the connection between physical activity and sedentary behavior of middle and high school students and mention the necessity of this study further. Also, I would like you to explain why Portugal, Italy, and Spain were selected among many European countries based on the analysis of previous studies.
Please clearly present the hypothesis of this study at the end of the introduction.
Method of research
Please explain the IPAQ's survey in detail in the research tool and add what questionnaire was used in this questionnaire.
Did you conduct the survey online through e-mail? Did you conduct the survey offline? Please add more information
Result
It is mentioned that the result value is not described at all and there are only differences between variables. Please derive the results by describing the difference value between variables. As described in Figure 2, please describe the contents of Table 2 and the results by comparing and explaining the contents of Figure 1.
discussion
Please provide an appropriate explanation for why the Spanish university student had the highest level of sedentary life. On the contrary, please discuss why this result was found compared to the results of previous studies in which the opposite result of the thesis No. 10 was drawn,
I don't feel the need for this sentence. Please correct and supplement it
It is judged to be a study of basic data that requires increasing the level and time of physical activity by identifying problems in sedentary life. However, the problems of the result description method should be supplemented, and please reinterpret the results of this study with more previous studies in the discussion.
Overall, I think it is a good study that can be meaningful and basic data. However, through a general review of the introduction, research method, results, and discussion, please revise and supplement it according to the level of healthcare journal.
Author Response
Review Report (Reviewer 2)
Introduction
Reviewer comment: Prior research on physical activity and sedentary behavior was analyzed, but it is judged by the analysis of college students. In this section, I believe that the development of the introduction can be much smoother if I talk about the connection between physical activity and sedentary behavior of middle and high school students and mention the necessity of this study further. Also, I would like you to explain why Portugal, Italy, and Spain were selected among many European countries based on the analysis of previous studies.
- Please clearly present the hypothesis of this study at the end of the introduction.
Review: Dear reviewer, thank you for your insightful comments. The exploratory literature review analysis presented in this paper intentionally focuses on studies that have observed physical activity behaviors in the context of university students, since this is the demographic group of our study. In this sense, it was essential to specify our analysis in studies that examined similar populations. We sought to provide a general and appropriate empirical framework for understanding the physical activity patterns of university students. Regarding second point, these three countries share some cultural similarities, which may be relevant when looking at trends in physical activity behavior. In addition, previous studies carried out in other regions of Europe highlight the unique patterns of physical activity levels, particularly among young adults, making them suitable for comparative analysis. Thus, our aim was to investigate whether such patterns persist in these three southern European countries, which have been little explored in this specific age group.
Reviewer comment: Please clearly present the hypothesis of this study at the end of the introduction.
Review: (ll.85-90). Based on the existing literature and our own experience, the central hypothesis of this study is that higher education students, while often complying with international PA guidelines, are still predominantly engaged in sedentary behaviors. Furthermore, comparisons between countries may reveal variations, potentially attributable to different national policies and levels of investment in promoting active lifestyles.
Method of research
Reviewer Comment: Please explain the IPAQ's survey in detail in the research tool and add what questionnaire was used in this questionnaire.
- Did you conduct the survey online through e-mail? Did you conduct the survey offline? Please add more information
Answer: Thanks for your comment. As requested, we make the changes proposed by the reviewer. We have completed the information found in the Instruments section.
Review: “The questionnaires were sent an online form (through e-mail), and each of the vali-dated versions for each country was administered to students in the respective country.” Page number 3, Line number 118-120.
Results
Reviewer Comment: It is mentioned that the result value is not described at all and there are only differences between variables. Please derive the results by describing the difference value between variables. As described in Figure 2, please describe the contents of Table 2 and the results by comparing and explaining the contents of Figure 1.
Answer: Thanks for your comment. As requested, we make the changes proposed by the reviewer. We have completed the information found in the Results section.
Review: “In regarding to Sitting (weak/day) we found differences between Portugal and Spain (p<.001, η2=0.164, Effect Size=0.885), Portugal and Italy (p<.001, η2=0.074, Effect Size=0.565), and Spain and Italy (p<.001, η2=0.236, Effect Size=1.113); Sitting (Weekend/day) between Portugal and Spain (p<.001, η2=0.064, Effect Size=0.524), Portugal and Italy (p<.001, η2=0.142, Effect Size=0.813), and Spain and Italy (p<.001, η2=0.205, Effect Size=1.014); Walking between Portugal and Spain (p<.001, η2=0.146, Effect Size=0.828), Portugal and Italy (p=.011, η2=0.054, Effect Size=0.476), and Spain and Italy (p<.001, η2=0.091, Effect Size=0.633); Moderate Activity between Portugal and Italy (p<.001, η2=0.074, Effect Size=0.567) and Spain and Italy (p=.011, η2=0.049, Effect Size=0.453); Vigorous Activity between Portugal and Italy (p<.001, η2=0.118, Effect Size=0.732) and Spain and Italy (p<.001, η2=0.076, Effect Size=0.573).” Page number 5, Line number 188-196.
Discussion
Reviewer comment: Please provide an appropriate explanation for why the Spanish university student had the highest level of sedentary life. On the contrary, please discuss why this result was found compared to the results of previous studies in which the opposite result of the thesis No. 10 was drawn.
Answer: Thanks for your comment. As requested, we make the changes proposed by the reviewer. “Previous studies show that the main barrier for Spanish university students is that they do not like to practice physical activity. This barrier has a high negative correlation with the levels of physical activity. A recent study shows that less than half of Spanish university students can be considered physically active.” (with references, lines 176-180).
Reviewer comments: It is judged to be a study of basic data that requires increasing the level and time of physical activity by identifying problems in sedentary life. However, the problems of the result description method should be supplemented, and please reinterpret the results of this study with more previous studies in the discussion.
Answer: Thanks for your comment. As requested, we make the changes proposed by the reviewer by adding some references in lines 177, 178, 200 and 215.

Round 2
Reviewer 1 Report
Comments and Suggestions for Authors
Dear Authors,
Congratulations on your efforts in correcting the manuscript, which is now considerably better with the improvements presented. Based on the new version presented, I have reached my decision about the manuscript, which will be forwarded to the Editor.
Sincerely,
Reviewer 2 Report
Comments and Suggestions for Authors The request for revision has been well reflected. Please review the final version more carefully to increase the degree of completion.